# Descriptive Study of Gut Microbiota in Infected and Colonized Subjects by *Clostridiodes difficile*

**DOI:** 10.3390/microorganisms9081727

**Published:** 2021-08-13

**Authors:** Pedro Sánchez-Pellicer, Vicente Navarro-López, Ruth González-Tamayo, Coral Llopis-Ruiz, Eva Núñez-Delegido, Beatriz Ruzafa-Costas, Laura Navarro-Moratalla, Juan Agüera-Santos

**Affiliations:** 1MiBioPath Group, Health and Science Faculty, Catholic University of Murcia, Campus de los Jerónimos, 135, 30107 Murcia, Spain; eva.nunez@bioithas.com (E.N.-D.); beatriz.ruzafa@bioithas.com (B.R.-C.); laura.navarro@bioithas.com (L.N.-M.); juan.aguera@bioithas.com (J.A.-S.); 2Infectious Diseases Unit, University Hospital of Vinalopó, Carrer Tonico Sansano Mora, 14, 03293 Elche, Spain; 3Biochemistry Laboratory, Vega Baja Hospital, Carretera Orihuela-Almoradí s/n, 03314 San Bartolomé, Spain; gonzalez_ruth@gva.es; 4Microbiology Laboratory, University Hospital of Vinalopó, Carrer Tonico Sansano Mora, 14, 03293 Elche, Spain; cllopis@vinaloposalud.com

**Keywords:** *Clostridiodes difficile*, *Clostridiodes difficile* infection, *Clostridiodes difficile* colonization, gut microbiota, 16S ribosomal RNA, dysbiosis

## Abstract

*Clostridiodes difficile* can lead to a range of situations from the absence of symptoms (colonization) to severe diarrhea (infection). Disruption of gut microbiota provides an ideal environment for infection to occur. Comparison of gut microbiota of infected and colonized subjects could provide relevant information on susceptible groups or protectors to the development of infection, since the presence of certain genera could be related to the inhibition of transition from a state of colonization to infection. Through high-throughput sequencing of *16S rDNA* gene, we performed alpha and beta diversity and composition studies on 15 infected patients (Group CDI), 15 colonized subjects (Group P), and 15 healthy controls (Group CTLR). A loss of alpha diversity and richness and a different structure have been evidenced in the CDI and P groups with respect to the CTRL group, but without significant differences between the first two. In CDI and P groups, there was a strong decrease in phylum Firmicutes and an expansion of potential pathogens. Likewise, there was a loss of inhibitory genus of *C. difficile* germination in infected patients that were partially conserved in colonized subjects. Therefore, infected and colonized subjects presented a gut microbiota that was completely different from that of healthy controls, although similar to each other. It is in composition where we found that colonized subjects, especially in minority genera, presented differences with respect to those infected.

## 1. Introduction

*Clostridioides difficile* is a Gram-positive bacillus that is strictly anaerobic and spore-forming [1]. It is one of the main causes of nosocomial diarrhea in hospitalized patients. Its pathogenicity is associated with the use of antibiotics and a decreased immune response, as well as with advanced age, hospitalization, and greater severity of underlying disease [2,3]. Clinical issues are due to production of the toxins TcdA and TcdB of cytotoxic action [4]. However, intestinal colonization of *C. difficile* can lead to a range of situations such as an absence of symptoms (colonized subjects) to severe diarrhea or fulminating pseudomembranous colitis (infected subjects) [5,6].

*C. difficile* infection (CDI) is an especially important problem in terms of mortality, morbidity, and associated costs. In addition, the risk of recurrence is extremely high. It has also been shown that the epidemiology of the CDI has changed since the beginning of the 21st century. Apart from an increase in mortality and morbidity generated by the CDI, existence of many cases from the community has begun to be described [7].

Disruption of endogenous gut microbiota (dysbacteriosis) provides the ideal environment for infection to occur; however, a healthy intestinal microbiota can prevent it. This phenomenon is called resistance to colonization [8]. A corrupted mechanism of resistance to colonization by *C. difficile* key is concern to the metabolism of bile acids [9]. It has been shown that primary bile acids have a germinating activity on spores of *C. difficile*, while secondary bile acids have an inhibitory activity on germination [10]. The gut microbiota provides enzymes responsible for transforming primary bile acids into secondary bile acids (mainly Bile Acid 7α Dehydroxylase or BaiCD.) [11]. The members of the gut microbiota that contribute these enzymes produce an inhibitory effect on germination of *C. difficile*. These genera are present in an extremely low quantity and their loss is associated with CDI [11,12]. Other characteristic alterations of gut microbiota of patients with CDI are a depletion of butyrate producing families such as Lachnospiraceae and Ruminococcaceae [13] and an increase of opportunistic pathogens, mainly from the phylum Proteobacteria [14].

On the other hand, the rate of carriers or colonized subjects by *C. difficile* is variable and dependent on age. It is higher in those subjects with certain concomitant situations that favor colonization as in patients with cystic fibrosis, with inflammatory bowel disease, or with repeated contact with hospital environments [15,16].

Comparison of gut microbiota of individuals infected and colonized by *C. difficile* has been minimally studied, and it is important to know what occurs in the gut microbiota of colonized subjects by *C. difficile* so that individuals do not develop clinical signs. This approach could provide relevant information on susceptible groups or protectors to the development of CDI. This approach will also provide information on pathogenetic mechanisms underlying CDI and to design more focused treatments (including probiotics to modulate gut microbiota) in the future.

Our hypothesis is that the gut microbiota of CDI patients and colonized subjects by *C. difficile* is quite different in terms of diversity, richness, structure, and composition with respect to healthy controls, but it is nonetheless quite similar to each other. The differences existing in individuals colonized and infected by *C. difficile* with respect to gut microbiota would be located in a set of dissimilarities in composition, which would explain, at least in part, the non-evolution from a state of colonization, which is observed in some cases, to a state of infection with the development of symptoms.

The main objective of this study is the comparison of the gut microbiota between a group of patients with CDI and a group of colonized subjects by *C. difficile* with respect to a group of healthy controls.

## 2. Materials and Methods

### 2.1. Study Design and Subject Inclusion

This is a pilot, descriptive, observational, cross-sectional study comparing the gut microbiota of 15 patients with CDI (Group CDI), 15 colonized subjects by *C. difficile*, (Group P) and 15 healthy controls (Group CTLR). The study subjects were included from the University Hospital of Vinalopó (Elche, Spain). The study was approved by the Ethics Committee for Clinical Research of the University Hospital of Vinalopó (Elche, Spain) on 9 November 2017.

A case of CDI was defined as one that presented diarrhea (3 stools or more cataloged from type 5 on the Bristol scale in the last 24 h), clinical diagnosis of pseudomembranous colitis, or toxic megacolon where the presence of toxigenic *C. difficile* was evidenced by the following laboratory tests [15]:

Detection of *C. difficile* TcdA and/or TcdB toxins and/or glutamate dehydrogenase (GDH) in feces by immunochromatography (*C. difficile GDH-toxins A-B MonlabTests*, Monlab, Barcelona, Spain). It is a quick test that enables the qualitative detection of TcdA, TcdB, and GHD simultaneously. Specific antibodies against TcdA, TcdB, and GDH are adsorbed in a stationary phase that consists of a membrane. If any of them are present in the stool sample, previously diluted, a specific reaction will occur when adding a conjugated antibody that will be manifested by a colored band.

Detection of the *tcdB* gene of *C. difficile* toxin TcdB in feces by PCR (*FluoroType CDiff*, Hain Lifescience, Nehren, Germany).

Detection of the *triose phosphate isomerase* (*TPI*) gene of *C. difficile* in feces by PCR (*FluoroType CDiff*, Hain Lifescience, Nehren, Germany). Procedure for detection of both *tcdB* and *TPI* genes consisted of two main steps. The first step was the extraction of DNA from stool samples using the GXT Stool Extraction kit, specifically adapted for the GenoXtract instrument (Hain Lifescience, Nehren, Germany). The second step is that extracted DNA were added to amplification and detection mixture that was composed of nucleotides, buffer, Taq polymerase, primers, and complementary DNA fluorescence-labeled for detection *TPI* and *tcdB* genes. The basis of the test is the quenching phenomenon. Once the mixture with the specific reagents was made, it was introduced into FluoroCycler instrument (Hain Lifescience, Nehren, Germany) for PCR amplification and hybridization. After this, the fluorophores were excited by light, emitting a fluorescence that was measured, while the temperature was raised gradually. If the *tcdB* and/or *TPI* genes were present in the sample, two peaks will appear in the melting curve.

For definition of a case of colonization by *C. difficile*, the subject has to present an absence of digestive symptoms or, in the case of having diarrhea, it had to be attributable to other causes, together with the laboratory tests mentioned for cases of CDI to demonstrate the presence of *C. difficile* in feces, taking into account that non-toxigenic strains (no detection of *tcdB* gene in feces by PCR and detection of *TPI* gene in feces by PCR) always were included in the colonized group. Regarding the healthy controls, they were volunteers without digestive symptoms, with negative tests for the detection of *C. difficile* in feces and without antibiotic administration in the last three months. Pediatric patients were not included in any of the study groups. The clinical and demographic characteristics of the study groups are summarized in Table 1.

### 2.2. Metataxonomic Determination of the Composition of the Gut Microbiota

Fecal samples for the study of the gut microbiota were collected and frozen −80 °C until their processing for the analysis.

For the isolation of bacterial DNA from fecal samples, the *MagNA Pure 2.0 LC* robot and the *MagNA Pure LC Total Nucleic Acid Isolation Kit - Large Volume* were used (Roche Diagnostics, Mannheim, Germany). To assess a correct extraction, we performed the quantification of the bacterial DNA obtained from each sample using the *Qubit Fluorometric Quantitation* reader using the *Qubit 1x dsDNA HS Assay kit* (Thermofisher, Massachusetts, MA, USA). 

To determine the composition of the gut microbiota, we performed high-throughput sequencing of the *16S rDNA* gene. The bacterial *16S rDNA* gene amplicons were obtained following the *16S rDNA gene Metagenomic Sequencing Library Preparation Illumina protocol* (Illumina, San Diego, CA, USA). The specific amplified sequences are in the V3 and V4 regions (459 bp) of the *16S rDNA* gene. The specific primers were selected from the current bibliography in this regard [17]:

Sequence forward primer: TCGTCGGCAGCGTCAGATGTGTATAAGAGACAGCCTACGGGNGGCWGCAG.

Sequence reverse primer: GTCTCGTGGGCTGGAGATGTGTATAAGAGACAGGACTACHVGGGTATCTAATCC.

The amplicon libraries were sequenced in a *MiSeq* Sequencer (Illumina, San Diego, CA, USA) following the manufacturer’s specifications.

### 2.3. Bioinformatic Analysis of the Amplicon Sequences of the Bacterial 16S rDNA Gene and Taxonomic Assignment

The bioinformatic analysis consisted of several stages. In the first place, a quality control of the obtained sequences and an elimination of the chimeric sequences were carried out, among other filtering processes. Then, it was possible to proceed with the taxonomic assignment.

For the metataxonomic analysis of the sequences, QIIME 2 (http:quiime.org, accessed on 17 January 2019) was used, which allowed us to perform the vast majority of the analyses necessary for the study of the microbiota through free access plugins, obtaining graphs and statistics. Through a pipeline designed specifically for this purpose, written in RStatistics environment (https://cran.r-project.org/index.html, accessed on 17 January 2019), all the steps were concatenated, and the workflow was generated from the moment we obtained the sequences to the report final.

Taxonomic assignment was carried out using the SILVA_release_132 database (https://www.arb-silva.de, accessed on 17 January 2019). For taxonomic affiliations, a naive Bayesian classifier integrated in QIIME 2 was used (https://github.com/qiime2/q2-feature-classifier, accessed on 17 January 2019). Each operational taxonomic unit (OTU) correlated by definition 97% with the sequence of the database.

Ecological diversity and richness studies were carried out using the Community Ecology Package plugin (https://cran.r-project.org/web/packages/vegan/index.html, accessed on 17 January 2019).

### 2.4. Expression of Results

Results that we obtained are expressed in their original form as relative abundance in percentages of the different phylum, class, order, family, and genus. To compare the structure of gut microbiota in study groups, we used concepts from microbial ecology such as diversity and richness.

For study of the alpha diversity, the Shannon and Simpson statistical indices and the ACE (abundance-based coverage estimator) and CHAO1 indices as richness estimators were used. To assess whether there were statistically significant differences at the genus level of the mean statistical indices of alpha diversity and richness of study groups, we used the non-parametric Wilcoxon test. The rarefaction curves were also constructed in the 3 study groups.

UniFrac model was used for the study of beta diversity. UniFrac distances were calculated using the Qiime2 plugin qiime2-qiime-diversity-tare-metrics-phylogenetic. Emperor viewer (http://emperor.microbio.me, accessed on 17 January 2019) was used to obtain the two-dimensional graphs of the 2 principal components corresponding to each axis.

To assess the differences in the composition of gut microbiota of the 3 study groups, we analyzed the mean of the relative abundances at the phylum, family, and genus levels and applied the non-parametric Wilcoxon test to assess statistically significant differences.

## 3. Results

### 3.1. Alpha Diversity in Study Groups

All alpha diversity indices (Shannon and Simpson) and richness estimators (ACE and CHAO1) showed statistically significant differences between the groups of infected and colonized subjects versus the group of healthy controls. However, there were no statistically significant differences between both groups of infected and colonized subjects. Table 2 shows the results of the comparison of the mean of these indices in each study group.

In the rarefaction curves we observed that all the samples reached the asymptotic region; therefore, the sampling was adequate, that is, the sequencing had sufficient depth since we obtained a number of OTUs representative at the genus level.

### 3.2. UniFrac Model

UniFrac analysis showed a principal component analysis that explained a variability of 42.60% (Principal Component 1, axis 1) and 14.95% (Principal Component 2, axis 2) (Figure 1).

In general, samples from a group of healthy controls formed a distinct cluster from the samples from infected and colonized subjects. Samples of infected and colonized subjects appeared in an overlapping manner but in a region in the biplot totally different from that of healthy controls. This implies that the structure of the gut microbiota of healthy controls is different from that of infected and colonized individuals. The infected and colonized samples did not form different clusters from each other.

### 3.3. Composition Analyses of Gut Microbiota

The main differences in the composition of gut microbiota in the three study groups in terms of phylum Firmicutes are summarized in Table 3.

We observed a statistically significant reduction in the phylum Firmicutes in infected and colonized subjects with respect to healthy controls, mainly at the expense of the reduction in genera such as *Agathobacter*, *Roseburia*, *Faecalibacterium*, *Ruminococcus*, and *Subdoligranolum* of the main butyrate-producing families such as Lachnospiraceae and Ruminococcaceae. The genus *Clostridiodes* (which includes only the species *Clostridiodes difficile*) had the same presence in infected and colonized subjects. Obviously, its presence was not detected in healthy controls. We also observed an increase with not much statistical significance (except *Enterococcus*) of lactic acid-producing bacteria, such as *Enterococcus*, *Streptococcus*, and *Lactobacillus*. We also highlight the increase of *Veillonella*, a microorganism considered opportunistic pathogen, with not much statistical significance, in infected and colonized subjects compared to healthy controls. 

Within the phylum Firmicutes, the differences in composition with respect to the genera that present enzymatic endowment to biotransform primary bile acids into secondary bile acids are summarized in Table 4. These genera would present an inhibitory capacity for the germination of *C. difficile* spores.

We generally observed a practical eradication of genus with enzymatic activity to biotransform primary bile acids into secondary bile acids in infected subjects with respect to healthy controls. In many genera, this decrease is statistically significant. We also observed this decrease in colonized subjects with respect to healthy controls. However, we consider that they are not pronounced, and we could speak of a partial conservation, with little statistical significance.

With respect to other variations in composition of genera of rest of phylum, they are summarized in Table 5.

We found statistically significant reduction in patients with CDI with respect to healthy controls in the phylum Actinobacteria, and on the other hand, there was a statistically significant increase in the phyla Bacteroidetes and Proteobacteria. Also noteworthy is the non-statistically significant increase in Verrucomicrobia in patients with CDI with respect to healthy controls. We also observed an increase in Bacteroidetes and Proteobacteria phyla statistically significantly in colonized subjects with respect to healthy controls. Also noteworthy is the reduction with not much statistical significance in the phyla Actinobacteria and Verrocomicrobia. Therefore, regarding the composition at the phylum level, there was a remarkable increase of Bacteroidetes and Proteobacteria in infected and colonized subjects compared to healthy controls, a practical eradicating of Actinobacteria in infected subjects and the partial conservation in colonized and an increase of Verrucomicrobia in infected subjects, and a decrease in colonization with respect to healthy controls. The increase in the phylum Bacteroidetes and the family Bacteroidaceae in the CDI and P groups was mainly at the expense of genus *Bacteroides*. The increase in the phylum Proteobacteria and the family Enterobacteriaceae in the CDI and P groups was mainly at the expense of genus *Escherichia-Shigella*. With respect to phylum Actinobacteria, at the genus level, there was a statistically significant decrease in *Bifidobacterium* in infected subjects with respect to healthy controls, while in colonized subjects, this decrease was not so marked. What happens in the phylum Verrucomicrobia is entirely at the expense of genus *Akkermansia*. Therefore, we observed a statistically significant increase in genus *Akkermansia* in infected subjects with respect to healthy controls and a decrease with less statistical significance in colonized subjects with respect to healthy controls.

## 4. Discussion

Results show a loss of alpha diversity and richness in infected and colonized subjects with respect to healthy controls. No statistically significant differences were observed between these two groups. This is a frequent finding in patients with CDI [18] and has also been observed in colonized subjects by *C. difficile* [19] and even in patients with nosocomial diarrhea not due to *C. difficile* [13]. This finding does not differentiate the state of colonization from that of infection. Administration of antibiotics has been considered one of the main triggers of dysbacteriosis with loss of alpha diversity, which is an essential risk factor for the development of CDI [20]. In addition, as has been observed in some of the few prospective studies in this regard, loss of alpha diversity is not an inexcusable development factor towards CDI [21]. In a group of patients with CDI, 93% (14 of 15) had received antibiotics in the previous 3 months, mainly cephalosporins, beta-lactamics, and fluoroquinolones, which are considered of high risk. In colonized subjects, the rate of previous antibiotic administration was 53% (8 of 15), although in the remaining 27%, these data were unknown (4 of 15). Therefore, we have a cause that largely explains the loss of alpha diversity in both groups. There are other factors different from administration of antibiotics that can lead to a loss of alpha diversity, such as liver disease, inflammatory bowel disease, and malignant blood diseases [22]. The presence of these situations occurred in 47% (7 of 15) of the patients in the CDI group, while they were not observed in any of the colonized individuals and healthy controls.

The statistical tool we used to assess beta diversity in gut microbiota of study groups was the UniFrac analysis, the objective of which is to assess whether there is any factor that explains most of the variability of data. We observed that structure of gut microbiota of healthy controls is completely different from that of infected and colonized subjects, which would overlap. Therefore, we could not use beta diversity as a notable differentiator of the state of colonization and infection by *C. difficile*. This finding is in accordance with the bibliography in this regard [19]. Another finding that we can extrapolate from the UniFrac model is the greater interindividual variability in infected and colonized subjects than in healthy controls where the samples are grouped in a more specific region. This finding is linked to the fact demonstrated in prospective studies that gut microbiota, in the context of infection or colonization by *C. difficile* in a hospital setting, is subject to a dynamic of changes [20], which would lead to the greater variability observed in these study groups. This factor has not been found to be significant in colonized subjects from the community setting, such as that which occurs in most of the subjects included in group P of the present study (87%); therefore, we do not know if this factor could influence this group. A total of 47% of subjects of CDI group came from the hospital environment compared to the remaining 53% that came from the community; therefore, this could have an influence in some way.

Loss of genera belonging to the main butyrogenic families Lachnospiraceae and Ruminococcacaceae in infected and colonized subjects is particularly significant and deep. This finding is very characteristic of patients with CDI [13,14] but has also been observed in colonized subjects [19] and in patients with nosocomial diarrhea not due to *C. difficile* [13]. Regarding the genera, at the expense of which the decreases in these families occur, greater variability appears in the bibliography. This may be since gut microbiota forms a dynamic ecosystem with physiological, metabolic, immunological, and protection functions against pathogens, which is stable on the basis that there is a degree of functional redundancy of its members, that is, of a core functionality. Therefore, the relative decrease in the genera of these majority families compared to cohorts of reference subjects can be variable since a few genera would exercise the same functions. However, in previous studies [13,14,19], there are decreases in genera that appear in this study such as *Faecalibacterium*, *Subdoligranulum*, and *Roseburia*. Regarding butyrate as a key metabolite in intestinal homeostasis, it only comes from bacterial anaerobic fermentation and is the colonocyte’s main source of energy. Butyrate is absorbed and oxidized, which reduces the osmotic load in the colon due to the presence of indigestible carbohydrates and, as the main anion in the colon, they form an acidic environment that would prevent the proliferation of intestinal pathogens such as *C. difficile*. In addition to these properties, butyrate is a powerful anti-inflammatory with the ability to regulate the secretion of pro-inflammatory cytokines in different intestinal immune cells through different mechanisms. They even have an influence on regulatory T lymphocytes, which decrease Th17 lymphocytes, whose activation stimulates granulopoiesis and the recruitment of neutrophils at the site of infection [23]. Therefore, we observed that in infected and colonized subjects a gut microbiota has been established with a very reduced capacity to synthesize butyrate, which would generate a pro-inflammatory environment but not enough to differentiate the state of colonization and infection.

Another important alteration in the composition within the phylum Firmicutes in this study was the greater conservation of genera with activity to biotransform primary bile acids into secondary bile acids in colonized subjects compared to infected subjects, where we have observed greater eradication, with moderate statistical significance (Table 4). The gut microbiota provides the enzyme BaiCD that transforms primary bile acids into secondary acids; however, this capacity is limited to a small group of bacteria [12]. Bacteria whose genome encodes these enzymes would produce an inhibitory effect on the germination of *C. difficile* spores. Losing these genera would mean a loss of this mechanism of resistance to colonization of intestinal pathogens. These specific species with BaiCd enzymatic capacity are mainly found in the genus *Clostridium* (cluster *Clostridium* XIVa), *Blautia*, and *Eubacterium* [12], although the activity is highly variable between species [24]. A recent study quantified the *BaiCD* gene cluster as a measure of the levels of intestinal bacteria with BaiCD activity, and a strong negative correlation with CDI was found [11]. The results of this study suggest that the relationship between bile acid metabolism by members of gut microbiota and its relationship with the pathogenesis of CDI is an essential mechanism, and the approach of the present study through a group of patients with CDI and a group of colonized subjects show this. However, it is suggested that in the future we will resort to a combination of metabolomic, functional genomics, and metagenomics studies, together with the study of the *16S rDNA* gene to consolidate this evidence. It is possible that, the total number of members of gut microbiota with BaiCD activity may be unknown. Metabolism of bile acids by gut microbiota differentiates the state of colonization from that of infection in this study.

We observed an increase in lactic acid-producing bacteria such as *Streptococcus*, *Enterococcus*, and *Lactobacillus* in infected and colonized subjects compared to healthy controls, with greater statistical significance in the case of *Enterococcus*. We consider this expansion as an "effect" and not as a "cause" of dysbacteriosis in infected and colonized subjects, because butyrate oxidation occurs in the colonocyte, mainly contributed by members of the phylum Firmicutes. Butyrate is transformed to CO_2_ by consuming O_2_. The dysbacteriosis observed in CDI and P groups would cause metabolic reorientation towards anaerobic glycolysis, lower O_2_ consumption, and increased oxygenation of the colonocyte surface, with the consequent expansion of facultative anaerobes such as these lactic acid-producing bacteria [25]. The literature has not emphasized much on the importance of CDI pathogenesis in these genera. *Streptococcus* and *Enterococcus* are opportunistic pathogenic bacteria, and *Lactobacillus* has potential probiotic effects. On the other hand, lactic acid in vitro reduces TcdA in a dose-dependent way and the bacterial load of *C. difficile* in a dose-independent way [26]. These genera present a duality of protective and predisposing effects of CDI and future studies should be used, for example to determine *Enterococcus* species and virulence factors such as cytosilin that are well established. However, presence of these genera does not differentiate the state of colonization from that of infection. Perhaps the genus that has been most affected that increases in patients with CDI is *Enterococcus* [13,14,27]. However, this increase has also been evidenced in colonized subjects [19,28].

Genus *Clostridiodes*, which is equivalent to *C. difficile* since it only contains that species, presented the same relative abundance in infected and colonized subjects compared to non-presence, as expected in healthy controls. Therefore, the load of *C. difficile* did not differentiate the state of colonization from that of infection in this study. Furthermore, the results show that the positivity of *tcdB* gene also does not distinguish between colonization and infection, since 53% of the group of colonized individuals presented toxigenic strains (Table 1). This is in line with other works in this regard [29]. In addition, it has also been seen that the toxin load of *tcdB* gene (colonized and infected subjects with high load *tcdB* gene *versus* low high *tcdB* gene) did not distinguish between colonization and infection [28]. Data of this study point to importance of evaluating virulence factors of *C. difficile* in the future, together with studies of *16S rDNA* gene and metagenomic studies. That is why factors concerning *C. difficile* should be considered in the differentiation between infection and colonization. For example, *tcdA* and *tcdB* genes are located at *PaLoc* or locus of pathogenicity. Interestingly, changes can occur in coding region of *PaLoc* as insertions, deletions, and point mutations that make up genetic heterogeneity, giving rise to several different toxinotypes. This means that there could be strains with different activity and specificity of their toxins with respect to the reference strain of *C. difficile* VPI 10463 [30]. Therefore, toxinotypes are important because they show functional properties of the *C. difficile* toxin variants, that is, greater or lesser activity and greater or less production. However, the correlation between the different toxinotypes that allows us to discriminate infection from colonization, as well as the severity of CDI, has not been clarified.

The increase of *Bacteroides* (and of phylum Bacteroidetes) in infected and colonized subjects could be surprising from the point of view that it is not a finding typically presented in previous studies, wherein decreases of this genus are normally observed [9,19]. However, some important studies have shown an increase in *Bacteroides* in patients with CDI compared to healthy controls [13]. Through murine models, an ability to mitigate CDI has been evidenced in some species [31] and it has an immunomodulatory activity in intestinal inflammatory processes that could limit the exacerbated immune response observed in patients with CDI [32]. On the other hand, *Bacteroides* has an unusual ability to recognize and metabolize a large quantity of polysaccharides from the diet and from the host itself. However, since competition for nutrients between members of gut microbiota is greater when groups of bacteria are more phylogenetically related [33], it would imply that the depletion of genus *Bacteroides* evidenced in patients with CDI and colonized subjects by many authors [9,19], although not observed in this study, could have less a priori influence on the loss of resistance to colonization of intestinal pathogens by nutrient competition mechanisms, since this would be more accused with loss of members of the phylum Firmicutes, especially of Clostridia class, which is phylogenetically closer to *C. difficile*.

An increase in Enterobacteriaceae at the expense of *Escherichia-Shigella* is another common finding in patients with CDI [14]. In this study, we also observed an increase in colonized subjects. This result has also been evidenced in recent studies [19,28]. We assume that the increase in *Escherichia-Shigella* occurs at the expense of *Escherichia coli* species in the majority, as has been evidenced by metagenomic studies [34]. There are different *E. coli*-producing diarrhea species such as enteropathogenic *E. coli*, enterotoxigenic *E. coli*, enteroinvasive *E. coli*, enteroaggregative *E. coli*, Shiga toxin-producing *E. coli*, verocytotoxigenic *E. coli*, and diffusely adherent *E. coli* [35]. Therefore, since similar pronounced increases of *Escherichia-Shigella* were found in patients with CDI and colonized subjects in this study, where *E. coli* would probably be a majority component of this cluster and knowing that there are several strains of enteropathogenic *E. coli*, it would be interesting to know the presence of these strains in both study groups. That is why we formulate the new hypothesis that there is a colonization of pathogenic strains in the CDI group that could enhance the effects of *C. difficile* toxins. On the other hand, this colonization would not occur in colonized subjects. This fact could partly explain the different clinical expressions in infection with respect to colonization of *C. difficile*.

The practical eradication of *Bifidobacterium* in infected subjects and the greater conservation in colonized subjects that we found in this study is an important point that differentiates the state of colonization and infection. Few microbiota studies have valued the importance of *Bifidobacterium* and its potential protective role in CDI. This genus has been seen to decrease in patients with CDI [14], and inconclusive results have been found in colonized subjects [19,28]. *Bifidobacterium* is a beneficial genus of antimicrobial and anti-inflammatory properties [36] whose decrease has been correlated with intestinal pathogen overgrowth [37], in vivo and in vitro inhibition of growth of *C. difficile* and reduced production [38] and neutralization of its toxins [39], and decreased tissue damage and mortality in infected mice [40]. We consider that a genus of beneficial properties of *Bifidobacterium* is eradicated in CDI patients and preserved in colonized subjects, an important result. This implies better control of *C. difficile* in colonized subjects. To our knowledge, it is the first time that this conclusion has been reached in microbiota studies.

The increase of *Akkermansia* in infected subjects and decrease in colonized subjects with respect to healthy controls is of special importance. The genus *Akkermansia* contains the species *Akkermansia muciniphila* (the only species isolated in humans) that has a highly effective capacity to ferment the mucin of the intestinal mucosa layer [41]. Regarding CDI, only two relatively recent studies have emphasized the importance that it could present in its pathogenesis. Sangster et al. observed an increase in *A. muciniphila* in 12 patients with CDI compared to 12 healthy controls. The authors highlighted that due to the ability of *A. muciniphila* to degrade mucin, and since *C. difficile* by itself is also capable of degrading mucin, it would provide it with a selective advantage of expanding, since it is able to adhere to a layer altered mucosa with better efficacy than other members of gut microbiota [27]. Another work that was published on the same date also evidenced an increase in *A. muciniphila* of 3.6% in patients with CDI, compared to 0.6% that was observed in subjects who did not receive antibiotic treatment. These authors pointed out that, although *A. muciniphila* has beneficial properties, its expansion in patients with CDI could be related to the modification of the intestinal microenvironment and could reflect the inflammation of the mucosa layer [14]. For first time, our results show an increase of *A. muciniphila* in patients with CDI and a decrease in colonized subjects. Since one of functions of the intestinal mucosa layer is protection against intestinal pathogens, an alteration in the integrity means that it is more permeable and allows greater access to the epithelium and this fact could generate inflammation. On the other hand, intestinal mucosa layer is also a potential source of nutrients for intestinal pathogens. This fact is evident in antibiotic treatment, since it disturbs gut microbiota and the availability of fucose and sialic acids in mucin, which facilitates expansion of *C. difficile* [42]. Therefore, the increase in patients with CDI reflects a greater degradation of intestinal mucosa and the decrease in colonized subjects would show a greater integrity. This finding differentiates the state of colonization from that of infection and would imply a greater control of *C. difficile* in colonized subjects.

We consider that the present study has two main limitations. The first limitation is the sample size. Each study group is made up of 15 subjects. This sample size has allowed us to find statistically significant differences in terms of diversity, richness, and composition in infected and colonized subjects with respect to healthy controls. However, the differences in the composition of colonized subjects with respect to those infected are in many cases not statistically significant. We think that with a larger sample size the differences discussed would have greater statistical significance due to the low statistical power of this study. The main obstacle to increasing the sample size has been the inclusion of colonized subjects by *C. difficile*, since they have been difficult to include and locate. The second limitation that we consider for interpretation of the results is diet. Diet is a factor that modulates the composition of gut microbiota, and in this study, no variables have been collected in this regard. We also think that this could have a greater influence on the gut microbiota of healthy controls. 

## 5. Conclusions

On the basis of the main objective of the study, which was the comparison of gut microbiota of patients with CDI and colonized subjects by *C. difficile* with respect to a group of healthy controls, we can conclude that infected and colonized subjects present a gut microbiota with a diversity, richness, structure, and composition completely different from that of healthy controls. However, gut microbiota of infected and colonized subjects showed great similarities in terms of diversity, richness, and structure. It is in composition where we find that colonized subjects, especially in minority genera, present differences with respect to those infected. This fact explains, at least in part, the state of colonization by *C. difficile.*

## Figures and Tables

**Figure 1 microorganisms-09-01727-f001:**
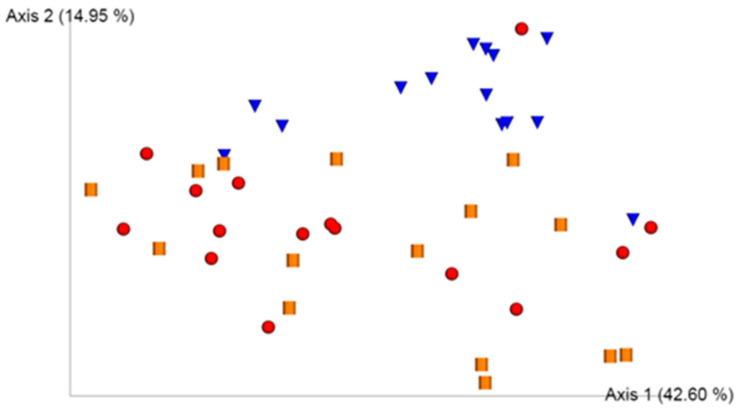
UniFrac model for the study groups. UniFrac model for the three study groups. Samples appear as red spheres (Group CDI), inverted blue cones (Group CTRL), and yellow cylinders (Group P).

**Table 1 microorganisms-09-01727-t001:** Clinical and demographic characteristics of the study groups.

Clinical and Demographic Characteristics	Group CDI	Group P	Group CTRL
Sex, number (%)			
Men	4 (26%)	10 (66%)	7 (46%)
Women	11 (74%)	5 (34%)	8 (54%)
Age (mean ± SD)	69 ± 19	51 ± 26	44 ± 12
Antibiotics last three months, number (%)	14 (93%)	8 (53%)	-
Cephalosporins	5 (33%)	2 (13%)	-
Fluorquinolones	4 (27%)	2 (13%)	-
Β-Lactamics	5 (33%)	3 (20%)	-
Others	5 (33%)	4 (27%)	-
Without antibiotics	1 (7%)	2 (13%)	-
Unknown	0 (0%)	4 (27%)	-
Strain Type, number (%)			
Toxigenic	15 (100%)	8 (53%)	-
Non-toxigenic	0 (0%)	7 (47%)	-
Comorbidities			
Hepatic disease	1 (7%)	0 (0%)	-
Crohn’s disease	1 (7%)	0 (0%)	-
Malignant blood disease	2 (13%)	0 (0%)	-
Other intestinal disease	3 (20%)	0 (0%)	-
Other comorbidity	13 (87%)	6 (40%)	-
Previous CD, number (%)	2 (13%)	0 (0%)	-
Origin, number (%)			
Hospital	7 (47%)	2 (13%)	-
Community	8 (53%)	13 (87%)	-
Resolution, number (%)			
Complete	11 (73%)	-	-
Exitus letalis	3 (20%)	-	-
Recurrence	1 (7%)	-	-

Clinical and demographic characteristics of the study groups. Group CDI: Subjects infected by *C. difficile*, Group P: Subjects colonized by *C. difficile*, Group CTRL: Healthy controls. SD: Standard deviation.

**Table 2 microorganisms-09-01727-t002:** Comparison of alpha diversity indices and richness estimators in study groups.

Group 1	Group 2	Index	Mean Group 1	Mean Group 2	*p*-Values Wilcoxon Test
CDI	CTRL	Shannon	2.0	2.8	0.0002
CDI	P	Shannon	2.0	1.9	0.3724
CTRL	P	Shannon	2.8	1.9	0.0003
CDI	CTRL	Simpson	0.7	0.9	0.0006
CDI	P	Simpson	0.7	0.7	0.3091
CTRL	P	Simpson	0.9	0.7	0.0003
CDI	CTRL	ACE	47.2	113.1	<0.0001
CDI	P	ACE	47.2	51.6	0.9339
CTRL	P	ACE	113.1	51.6	<0.0001
CDI	CTRL	CHAO1	47.2	113.1	<0.0001
CDI	P	CHAO1	47.2	51.6	0.9339
CTRL	P	CHAO1	113.1	51.6	<0.0001

**Table 3 microorganisms-09-01727-t003:** Composition analyses of key OTUs of phylum Firmicutes in study groups.

Family	Genus	Group CDI	Group P	Group CTRL	CDI versus CTRL	*P* versus CTRL	CDI versus *P*
		38.5533	39.1305	66.8691	0.0006	0.0016	1.0000
Lachnospiraceae		11.7971	12.8014	18.7516	0.0771	0.1213	0.8035
Lachnospiraceae	*Agathobacter*	0.0374	0.1254	6.2873	<0.0001	<0.0001	1.0000
Lachnospiraceae	*Roseburia*	0.1196	0.1654	2.7259	<0.0001	0.0001	0.6545
Ruminococcaceae		9.3575	6.3853	31.4165	<0.0001	<0.0001	0.1150
Ruminococcaceae	*Faecalibaterium*	3.2948	0.9126	11.2857	0.0013	<0.0001	0.1299
Ruminococcaceae	*Ruminococcus*	0.1022	0.0560	4.5458	0.0009	0.0002	0.3894
Ruminococcaceae	*Subdoligranolum*	0.3407	0.2808	0.2808	0.0001	<0.0001	0.4092
Peptostreptococcaceae		0.5298	0.4781	0.0600	0.0134	0.0242	0.5755
Peptostreptococcaceae	*Clostridiodes*	0.3668	0.3668	0.0000	<0.0001	<0.0001	0.1568
Enterococcaceae		0.6853	0.4993	0.0000	0.0009	0.0021	1.0000
Enterococcaceae	*Enterococcus*	0.6853	0.4993	0.0000	0.0009	0.0021	1.0000
Streptococcaceae		2.3442	3.5860	0.7527	0.0701	0.3481	0.1300
Streptococcaceae	*Streptococcus*	2.3431	3.5834	0.7448	0.0701	0.3481	0.1408
Lactobacillaceae		1.0287	0.2269	0.0111	0.3307	0.1475	0.8470
Lactobacillaceae	*Lactobacillus*	1.0017	0.2256	0.0111	0.3307	0.2624	1.0000
Veillonellaceae		3.4787	12.1379	10.2983	0.2216	0.7765	0.6185
Veillonellaceae	*Veillonella*	2.9121	4.5646	0.0824	0.1075	0.3898	0.8342

Composition analyses of key OTUs of phylum Firmicutes in study groups. Columns Group CDI, Group P, and Group CTRL are the mean of relative abundance (%). Columns CDI versus CTRL, *P* versus CTRL, and CDI versus *P* are the *p*-value of the Wilcoxon test.

**Table 4 microorganisms-09-01727-t004:** Composition analyses of genera of phylum Firmicutes with activity BaiCD in study groups.

Family	Genus	Group CDI	Group P	Group CTRL	CDI versus CTRL	P versus CTRL	CDI versus P
Lachnospiraceae	*Blautia*	0.5904	1.5385	0.6994	0.1831	0.1829	0.7395
Lachnospiraceae	*Eubacterium ventriosum*	0.0000	0.0032	0.7224	<0.0001	0.0001	0.3506
Lachnospiraceae	*Eubacterium eligens*	0.0063	0.1217	0.3189	0.0260	0.1126	0.5383
Lachnospiraceae	*Eubacterium xylanophilum*	0.0000	0.0010	0.1578	0.0001	0.0003	1.0000
Lachnospiraceae	*Eubacterium ruminantium*	0.0000	0.0000	0.0420	0.1498	0.1498	1.0000
Lachnospiraceae	*Eubacterium fissicatena*	0.0082	0.0485	0.0053	0.5155	0.6324	0.2879
Lachnospiraceae	*Eubacterium hallii*	0.0011	0.0056	0.0040	0.0740	0.2220	0.5772
Lachnospiraceae	*Eubacterium oxidoreducens*	0.0000	0.0000	0.0001	0.3340	0.3340	1.0000
Eubacteriaceae	*Eubacterium*	0.0011	0.0118	0.0003	0.5634	0.0790	0.2401
Clostridiales family XIII	*Eubacterium brachy*	0.0000	0.0051	0.3334	0.0004	0.0036	0.3506
Clostridiales family XIII	*Eubacterium nodatum*	0.0239	0.1099	0.0196	0.0261	0.1174	0.0040
Ruminococcaceae	*Eubacterium coprostanoligenes*	0.0939	0.1774	1.3705	0.0001	0.0001	0.9617

Composition analyses of genera of phylum Firmicutes with activity BaiCD in study groups. Columns Group CDI, Group P, and Group CTRL are the mean of relative abundance (%). Columns CDI versus CTRL, *P* versus CTRL, and CDI versus *P* are the *p*-value of the Wilcoxon test.

**Table 5 microorganisms-09-01727-t005:** Composition analyses of key OTUs in the remaining phyla in the study groups.

Phylum	Family	Genus	Group CDI	Group P	Group CTRL	CDI versus CTRL	P versus CTRL	CDI versus P
Bacteroidetes			39.3656	36.0149	19.5638	0.0275	0.0521	0.6783
	Bacteroidaceae		26.8470	29.1252	11.0028	0.0307	0.0137	0.7400
	Bacteroidaceae	*Bacteroides*	26.8470	29.1252	11.0028	0.0307	0.0137	0.7400
Protebacteria			14.3918	20.5514	3.8165	0.0154	0.0636	0.8357
	Enterobacteriaceae		10.6111	14.7078	3.0008	0.0521	0.3698	0.7399
	Enterobacteriaceae	*Escherichia-Shigella*	9.1685	10.6014	2.5561	0.1620	0.2051	1.0000
Actinobacteria			1.1694	2.7769	4.6633	0.0032	0.3261	0.0889
	Bifidobacteriaceae		0.6280	2.0634	3.7001	0.0076	0.3698	0.1653
	Bifidobacteriaceae	*Bifidobacterium*	0.6250	2.0620	3.6980	0.0071	0.3545	0.1564
Verrocomicrobia			5.9703	1.2742	4.7534	0.1605	0.0516	0.8698
	Akkermansiaceae		5.9703	1.2742	4.7534	0.1605	0.0516	0.8698
	Akkermansiaciae	*Akkermansia*	5.9703	1.2742	4.7534	0.1605	0.0516	0.8698

Composition analyses of key OTUs the remaining phyla in study groups. Columns Group CDI, Group P, and Group CTRL are the mean of relative abundance (%). Columns CDI versus CTRL, *P* versus CTRL, and CDI versus *P* are the *p*-value of the Wilcoxon test.

## Data Availability

The data presented in this study are available on request from the corresponding author. The data are not publicly available due to privacy restrictions.

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
