# Peer review of "Descriptive Study of Gut Microbiota in Infected and Colonized Subjects by Clostridiodes difficile"

_microorganisms, 2021, doi:10.3390/microorganisms9081727_

Round 1

Reviewer 1 Report

  1. Information on immunochromatography and PCR detection of C. difficile should be added to the Materials and methods section.
  2. Both group CDI and group P include patients with C. difficile. Does the number of C. difficile differ in the groups? Did you quantify C. difficile number with qPCR?
  3. Table 1. Why is the total percentage in the Comorbidities column greater than 100?

  4. Line 123: To assess "correct extraction" you should use several isolation methods. Qubit just quantifies DNA.
  5. Among comorbidities are intestinal diseases accompanied by significant changes in microbiota. Did you account for their contribution to changing bacterial diversity in studied groups?

  6. Figure 1-4: the text in the pictures is small and difficult to read even when zoomed in to see the details.

Author Response

Dear reviewer.

Thank you very much for your input. Next, we will comment on your comments and suggestions.

  • Information on immunochromatography and PCR detection of C. difficile should be added to the Materials and methods section: We add the requested information regarding the immunochromatographic and PCR methods.
  • Both group CDI and group P include patients with C. difficile. Does the number of C. difficile differ in the groups? Did you quantify C. difficile number with qPCR? No, we did not previously quantify difficile by qPCR, we only detected it qualitatively in patients with CDI and in colonized subjects. Surprisingly, our results show that by high- throughput sequencing of the 16S rDNA gene, C. difficile presented the same relative abundance in infected and colonized subjects. Therefore, the greater load of C. difficile does not differentiate the state of clinical infection from that of asymptomatic colonization.
  • Table 1. Why is the total percentage in the Comorbidities column greater than 100? It is possible because each study subject may present more than one comorbidity.
  • Line 123: To assess "correct extraction" you should use several isolation methods. Qubit just quantifies DNA. Exactly, Qubit is not a DNA extraction method. Qubit quantifies the amount of bacterial DNA extracted. For the extraction o isolation of bacterial DNA from fecal simples we use the MagNA Pure 2.0 LC robot, and the MagNA Pure LC Total Nucleic Acid Isolation Kit - Large Volume were used (Roche Diagnostics, Mannheim, Germany). We change the word extraction to isolation to avoid confusión.
  • Among comorbidities are intestinal diseases accompanied by significant changes in microbiota. Did you account for their contribution to changing bacterial diversity in studied groups? Indeed, there are other factors different from administration of antibiotics, which can lead to a loss of alpha diversity, such as, liver disease, inflammatory bowel disease and blood diseases malignant (Kriss M, Hazleton KZ, Nusbacher NM, Martin CG, Lozupone CA. Low diversity gut microbiota dysbiosis: drivers, functional implications and recovery. Curr OpinMicrobiol. 2018; 44: 34-40). The presence of these situations occurred in 47% (7 of 15) of the patients in the CDI group, while it was not observed in any of the colonized individuals and health controls. We add some sentences about this point in the paragraph where we talk about alpha diversity in the discusión.
  • Figure 1-4: the text in the pictures is small and difficult to read even when zoomed in to see the details. We decided to remove these figures 1, 2, 3 and 4 because we cannot obtain them at a higher resolution. also no more information than table 2.

Reviewer 2 Report

I red manuscript entitled "Descriptive study of gut microbiota in infected and colonized subjects by Clostridiodes difficile" with great interest, especially because in this study Authors compare intestinal microbiota of CDI patients, patients colonized with C. difficile and healthy subjects. This topic is under interest of many scientists, since the presence of certain genera could be related to the inhibition of transition from a state of colonization to infection. In this study Authors used appropriate and modern methodology:   high-throughput sequencing of 16S rDNA gene, alpha and beta diversity and composition studies were performed on 15 infected patients (Group CDI), 15 colonized subjects (Group P) and 15 healthy controls (Group CTLR). A loss of alpha diversity and richness and a different structure have been evidenced in the CDI and P groups with respect to the CTRL group, but without significant differences between the first two. In CDI and P groups there was a strong decrease in phylum Firmicutes and an expansion of potential pathogens. Authors observed a loss of inhibitory genus of C. difficile germination in infected patients that are partially conserved in colonized subjects. Especially, very interesting conclusion was made about Bifidobacterium, which was eradicated in CDI patients and preserved in colonized subjects. Also important  conclusion was made about Akkermansia: the increase of Akkermansia muciniphila in infected subjects and decrease in colonized subjects with respect to healthy controls. For first time, Authors  show an increase of A. muciniphila in patients with CDI and a decrease in colonized subjects.

I think that this paper is valuable and appropriate for publication in Microorganisms.

I'll advise Authors to pay attention on lines:

  1. line 67 - font size
  2. line 100 - tcdB gene - for gene names use Italic
  3. line 102 - as above
  4. Table 1 - because comorbitidies dominated in CDI and P groups, please explain them in the legend of Table, or in the text
  5. Figure 5 (line 202) in title are mentioned green cylinders (Group P), but color is yellow
  6. line 230 - font size
  7. line 348 - remove dot before "may be unknown"
  8. line 378 -  tcdB gene - for gene names use Italic

Sincerely

Prof. Gayane Martirosian M.D., Ph.D.

Department of Medical Microbiology Medical University of Silesia in Katowice, Poland

Author Response

Reviewer 2:

  • line 67 - font size: sorry but we do not understand this point.
  • line 100 - tcdB gene - for gene names use Italic: Corrected in all text.
  • line 102 - as above: Corrected in all text.
  • Table 1 - because comorbitidies dominated in CDI and P groups, please explain them in the legend of Table, or in the text: Please look at this reply to reviewer 1 “Among comorbidities are intestinal diseases accompanied by significant changes in microbiota. Did you account for their contribution to changing bacterial diversity in studied groups? Indeed, there are other factors different from administration of antibiotics, which can lead to a loss of alpha diversity, such as, liver disease, inflammatory bowel disease and blood diseases malignant (Kriss M, Hazleton KZ, Nusbacher NM, Martin CG, Lozupone CA. Low diversity gut microbiota dysbiosis: drivers, functional implications and recovery. Curr OpinMicrobiol. 2018; 44: 34-40). The presence of these situations occurred in 47% (7 of 15) of the patients in the CDI group, while it was not observed in any of the colonized individuals and health controls. We add some sentences about this point in the paragraph where we talk about alpha diversity in the discusión”.
  • Figure 5 (line 202) in title are mentioned green cylinders (Group P), but color is yellow:
  • line 230 - font size: We do not understand this point.
  • line 348 - remove dot before "may be unknown":
  • line 378 - tcdB gene - for gene names use Italic: Corrected in all text.
